# GODA: Goal-conditioned Data Augmentation

## Abstract

Offline reinforcement learning (RL) enables policy learning from pre-collected offline datasets, relaxing the need to interact directly with the environment. However, limited by the quality of offline datasets, it generally fails to learn well-qualified policies in suboptimal datasets. To address datasets with insufficient optimal demonstrations, we introduce *Goal-cOnditioned Data Augmentation* (GODA), a novel goal-conditioned diffusion-based method for augmenting samples with higher quality. Leveraging recent advancements in generative modeling, GODA incorporates a return-oriented goal condition with various selection mechanisms. Specifically, we introduce a controllable scaling technique to provide enhanced return-based guidance during data sampling. GODA learns a comprehensive distribution representation of the original offline datasets while generating new data with selectively higher-return goals, thereby maximizing the utility of limited optimal demonstrations. Furthermore, we propose a novel adaptive gated conditioning method for processing noised inputs and conditions, enhancing the capture of goal-oriented guidance. We conduct experiments on the D4RL benchmark and real-world challenges, specifically traffic signal control (TSC) tasks, to demonstrate GODA's effectiveness in enhancing data quality and superior performance compared to state-of-the-art data augmentation methods across various offline RL algorithms. Our code will be publicly accessible upon review.

## 1 Introduction

Reinforcement learning Sutton & Barto (2018) aims to learn a control policy from trial and error through interacting with the environment. While demonstrating remarkable performance in various domains, this approach typically requires vast amounts of training data collected from these interactions. Such data-intensive requirements become impractical in applications where environmental interactions are costly, risky, or time-consuming, such as robotics, autonomous driving Li et al. (2024) and traffic signal control (TSC) Zhang & Deng (2023). Offline RL offers a feasible solution to these challenges by enabling policy learning directly from pre-collected historical datasets, thus significantly reducing the need to interact directly with the environment.

Although offline RL makes policy learning less expensive, its performance is highly dependent on the quality of the pre-collected datasets and may suffer from lack of diversity, behavior policy bias, distributional shift, and suboptimal demonstrations Prudencio et al. (2023). The performance of offline RL tends to decline drastically when trained with suboptimal offline datasets. Previous studies have attempted to address these issues by constraining the learned policy to align closely with the behavior policy Lyu et al. (2022) or by limiting out-of-distribution action values Kostrikov et al. (2021). Although these approaches have shown performance improvements, they retain the inherent defects of offline datasets, remaining highly dependent on data quality.

Several studies have addressed the limitations of offline RL using data augmentation methods to generate more diverse samples. One approach involves learning world models to mimic environmental dynamics and iteratively generate synthetic rollouts from a start state Zhang et al. (2023). While this method significantly improves sample efficiency and data diversity, it suffers from compounding errors and fails to control the quality of generated trajectories. Other research leverages generative models to capture the distributions of collected datasets and randomly sample new transition data Lu et al. (2024). Although these methods demonstrate some performance improvements, they remain inefficient when dealing with datasets containing limited optimal demonstrations. This inefficiency stems from their inability to effectively control the quality of generated data.

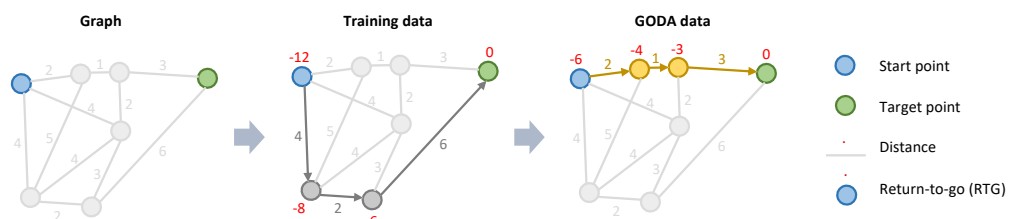

Figure 1: Illustrative examples of how GODA augments higher-return data with goal guidance. GODA utilizes scalable RTG-based goal conditions to generate samples with higher returns (shorter overall distance).

We attempt to address this challenge by taking advantage of generative modeling to augment higher-quality data with directional goals. Unlike previous studies that sample data unconditionally and randomly Lu et al. (2024), we introduce GODA to incorporate representative goals, guiding the samples toward higher returns. Given the exceptional performance of diffusion models Ho et al. (2020); Karras et al. (2022) in the field of generative artificial intelligence, GODA utilizes a diffusion model as its generative framework. GODA is trained to capture a comprehensive representation of the data distribution from the original dataset while sampling new data conditioned on selective high-return goals. This approach maximizes the utility of the limited well-performed trajectories in the original datasets. Inspired by Decision Transformer Chen et al. (2021), we define the 'goal' as the *return-to-go* (RTG), which represents the cumulative rewards from the current step until the end, coupled with its specific timestep in trajectories. RTG explicitly indicates the expected future rewards for a given behavior at the current timestep for a specific trajectory. We assume that at the same timestep across different trajectories, a higher RTG signifies a higher goal. To generate samples that exceed the quality of the original dataset, we introduce three *goal selection mechanisms* and a *scaling technique* to control our expected goals during sampling.

We present an illustrative example of how GODA operates in Figure 1. The task is to identify the shortest path from the starting point to the target. By setting higher RTG goals during sampling, GODA can potentially discover a more efficient route that yields a higher return (the RTG at the first timestep is equal to the return). To better incorporate goal conditions, we further propose a novel *adaptive gated conditioning* approach. This method utilizes a condition-adaptive gated residual connection and a gated long skip connection to capture multi-granularity information effectively. GODA is an off-the-shelf solution that can seamlessly integrate with other offline RL optimization approaches on various tasks to achieve superior results. We summarize our contributions:

**1)** We propose a goal-conditioned data augmentation method, namely GODA, for offline RL. It achieved enhanced data diversity and quality for offline datasets with limited optimal demonstrations. **2)** We introduce novel directional goals with selection mechanisms and controllable scaling to provide higher-return guidance for the data sampling process in our employed generative models. Additionally, we propose a novel adaptive gated conditioning approach to better capture goal guidance. **3)** We show GODA's competence through comprehensive experiments on the D4RL benchmark compared with state-of-the-art data augmentation methods across multiple offline RL algorithms. We further evaluate the effectiveness of GODA on a real-world application, i.e., traffic signal control with datasets obtained from widely used controllers in real-world deployments. These evaluations verify GODA's effectiveness in addressing various challenges, significantly enhancing the applicability of RL-based methods for real-world scenarios.

## 2 RELATED WORK

**Offline RL.** Conventional offline RL methods aim to alleviate the distributional shift problem, i.e., a significant drop in performance due to deviations between learned policy and the behavior policy used for generating the offline data Hu et al. (2023). To address this issue, various strategies have been employed, including explicit correction Xu et al. (2022), such as constraining the policy to a restricted action space Kumar et al. (2019), and making conservative estimates of the value function Yu et al. (2021); Kumar et al. (2020), with the aim of aligning the behavior policy with the learned

policy. Some recent studies exploit the strong sequence modeling ability of Transformer models to solve offline RL with trajectory optimization. For instance, Decision Transformer Chen et al. (2021) and its variants Wu et al. (2024); Gao et al. (2024) utilize a GPT model to autoregressive predict actions given the recent subtrajectories composed of historical RTGs, states, and actions. Diffusion models have also been adopted in offline RL given its exceptional capability of multi-modal distribution modeling. Diffuser Janner et al. (2022) employs diffusion models for long-horizon planning, effectively bypassing the compounding errors associated with classical model-based RL. Hierarchical Diffuser Chen et al. (2024) enhances this approach by introducing a hierarchical structure, specifically a jumpy planning method, to improve planning effectiveness further.

**Data augmentation in Offline RL.** Data augmentation proactively generates more diverse data to improve policy optimization. For instance, TATU Zhang et al. (2023) uses world models to produce synthetic trajectories and truncates those with high accumulated uncertainty. However, model-based RL often suffers from compounding errors in the learned world models. GuDA Corrado et al. (2024) introduces human guidance into data augmentation functions (DAFs), i.e., translation, rotation, and reflection, for generating expert-quality data, while human intervention is costly and lacks scalability. Diffusion models are also directly applied to data augmentation through the sampling process. SynthER Lu et al. (2024) is the first work that employs diffusion models to learn the distribution of initial offline datasets and unconditionally augment large amounts of new random data. However, it fails to control the sampling process to steer toward high-return directions actively. DiffStitch Li et al. (2024) attempts to enhance the quality of generated data by actively connecting low-reward trajectories to high-reward ones using a stitching technique.

We propose enhancing the quality of generated data from a different perspective by introducing a controllable directional goal into our generative modeling. This approach selectively reuses optimal trajectories to guide the sampling process toward achieving higher returns.

## 3 PRELIMINARIES

### 3.1 OFFLINE REINFORCEMENT LEARNING

In RL, the task environment is generally formulated as a Markov decision process (MDP) $\{\mathcal{S}, \mathcal{A}, \mathcal{S}, \mathcal{P}, \gamma\}$ Sutton & Barto (2018). $s \in \mathcal{S}$, $s' \in \mathcal{S}$, $a \in \mathcal{A}$, $r = \mathcal{R}(s, a)$, $\mathcal{P}(s'|s, a)$, and $\gamma \in [0, 1)$ represent state, next state, action, reward function, state transition, and discount factor, respectively. RL aims to train an agent to interact with the environment and learn a policy $\pi$ from experience. The objective of RL is to maximize the expected discounted cumulative rewards over time: $J = \mathbb{E}_\pi \left[ \sum_{t=0}^{\infty} \gamma^t \mathcal{R}(s_t, a_t) \right]$, where $t$ denotes the timestep in a trajectory. For offline RL, the policy is learned directly from offline datasets pre-collected by other behavior policies, instead of environmental interactions. The offline dataset typically consists of historical experience described as tuples $(s, a, r, s')$ and other environmental signals. After learning a policy $\pi(\mathcal{D})$ from dataset $\mathcal{D}$, the performance is evaluated in online environment as $\mathbb{E}_{\pi(\mathcal{D})} \left[ \sum_{t=0}^{\infty} \gamma^t \mathcal{R}(s_t, a_t) \right]$. While offline RL eliminates reliance on interacting with the environment, it is highly restricted by the quality of offline datasets due to the lack of feedback from the environment. Our GODA aims to enhance the diversity and quality of the dataset by upsampling the pre-collected data to an augmented dataset $\mathcal{D}^*$. The objective is to learn a policy $\pi(\mathcal{D}^*)$ that outperform $\pi(\mathcal{D})$ learned from original dataset $\mathcal{D}$, such that $\mathbb{E}_{\pi(\mathcal{D}^*)} \left[ \sum_{t=0}^{\infty} \gamma^t \mathcal{R}(s_t, a_t) \right] > \mathbb{E}_{\pi(\mathcal{D})} \left[ \sum_{t=0}^{\infty} \gamma^t \mathcal{R}(s_t, a_t) \right]$.

### 3.2 DIFFUSION MODELS

Diffusion models Sohl-Dickstein et al. (2015); Ho et al. (2020); Karras et al. (2022), a class of well-known generative modeling methods, aim to learn a comprehensive representation of the data distribution $p_{\text{data}}(\mathbf{x}^N)$ with a standard deviation $\sigma_{\text{data}}$ from a given dataset. Diffusion models generally have two primary processes, the *forward process*, also known as the *diffusion process*, and the *reverse/sampling process*. The forward process is characterized by a Markov chain in which the original data distribution $\mathbf{x}^N \in p_{\text{data}}(\mathbf{x}^N)$ is progressively perturbed with a predefined i.i.d. Gaussian noise schedule $\sigma^N = 0 < \sigma^{N-1} < \cdots < \sigma^0 = \sigma_{\text{max}}$. Therefore, we can obtain a sequence of noised distributions $p(\mathbf{x}^i; \sigma^i)$ for each nose level $\sigma^i$, where the last noised distribution $p(\mathbf{x}^0; \sigma_{\text{max}})$ can be seen as pure Guassion noise when $\sigma_{\text{max}} \gg \sigma_{\text{data}}$. Elucidated Diffusion Model (EDM) Karras et al. (2022) formulates the forward and reverse processes as a probability-flow ODE, where the

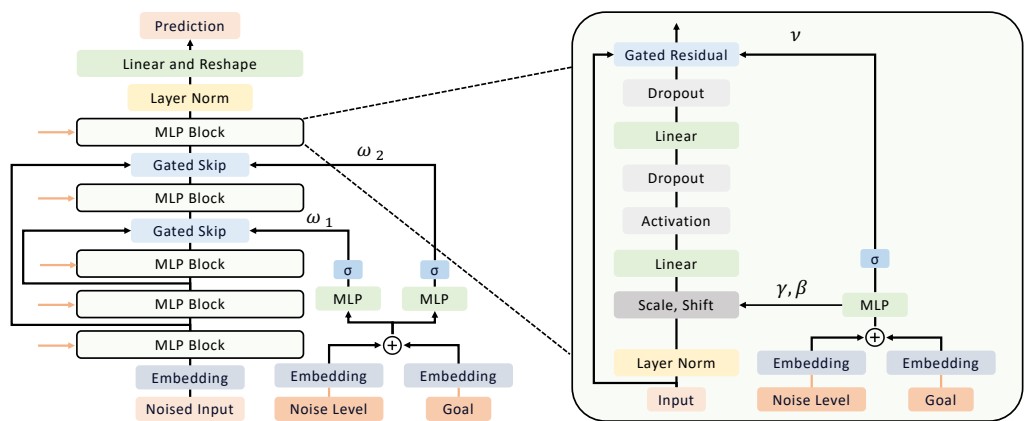

Figure 2: The denoiser neural network with adaptive gated conditioning architecture.

noise level can be increased or decreased by moving the ODE forward or backward in time:

$$\mathrm{d}\mathbf{x} = -\dot{\sigma}(t_i)\sigma(t_i)\nabla_{\mathbf{x}} \log p(\mathbf{x}; \sigma(t_i))\mathrm{d}t_i, \tag{1}$$

where $\dot{\sigma}(t_i)$ denotes derivative over denoise time and $\nabla_{\mathbf{x}} \log p(\mathbf{x}; \sigma(t_i))$ is referred to as the score function Song et al. (2020), which points towards regions of higher data density. It is worth noting that we use $t_i$ to denote the noise time to distinguish it from the trajectory timestep $t$. The ODE pushes the samples away from the data or closer to the data through infinitesimal forward or backward steps. The corresponding step sequence is $\{t_0, t_1, ... t_N\}$, where $t_N = 0$ and $N$ denotes the number of ODE solver iterations. EDM proposes to estimate the score function using denoising score matching Karras et al. (2022). Specifically, a denoiser neural network $D_\theta(\mathbf{x}; \sigma)$ is trained to approximate data $\mathbf{x}^N$ sampled from $p_{\text{data}}$ by minimizing the $L_2$ denoising loss independently for each $\sigma$:

$$\min_\theta \mathbb{E}_{\mathbf{x}^N \sim p_{\text{data}}; \mathbf{n} \sim \mathcal{N}(\mathbf{0}, \sigma^2 \mathbf{I})} \left\| D_\theta(\mathbf{x}^N + \mathbf{n}; \sigma) - \mathbf{x}^N \right\|_2^2. \tag{2}$$

Subsequently, the score function can be calculated as $\nabla_{\mathbf{x}} \log p(\mathbf{x}; \sigma) = (D_\theta(\mathbf{x}; \sigma) - \mathbf{x})/\sigma^2$.

## 4 METHODOLOGY

In this section, we introduce GODA, a goal-conditioned data augmentation method utilizing generative modeling for augmenting higher-quality synthetic transition data.

### 4.1 SELECTIVE GOALS AND CONTROLLABLE SCALING

#### 4.1.1 RETURN-ORIENTED GOAL

Prior diffusion-based work Lu et al. (2024) lacks the ability to guide the sampling process in the desired direction. We attempt to introduce a return-oriented goal as a condition of our diffusion model. Inspired by Decision Transformer, we adopt RTG Chen et al. (2021), cumulative rewards from the current step till the end, as the explicit goal condition $\hat{g}_t = \sum_{t'=t}^{T} r_{t'}$. For each transition sample represented as a tuple $(s, a, s', r)$ within a trajectory, RTG quantifies the expected future rewards for the current behavior, effectively serving as a goal. In other words, a higher RTG at a specific timestep typically signifies a higher goal for the policy to pursue. Since the same behavior at different timesteps often yields varying RTGs across different trajectories, we combine the RTG with its corresponding timestep in the trajectory as the condition for each specific transition sample. The timestep signal acts as a timestamp for each goal.

#### 4.1.2 SELECTIVE GOAL CONDITIONS

During dataset preprocessing, we first organize offline samples into trajectories, compute the RTG for each, and append timesteps to every sample. To fully leverage well-performing samples and

---

**Algorithm 1** GODA: Goal-Conditioned Data Augmentation

---

1: Initialize generative model $G_\theta$ and $\mathcal{D}^* = \emptyset$
2: Split initial offline dataset $\mathcal{D}$ into trajectories according to episode terminal information
3: Calculate RTGs for each transition sample in trajectories
4: Add RTGs and timesteps as goals into $\mathcal{D}$
5: Train $G_\theta$ on $\mathcal{D}$ using Eq. 6 by conditioning on goals
6: **repeat**
7:     Extract goal conditions for sampling according to the goal selection mechanism.
8:     Re-assign sampling goal conditions with goal scaling factor $\lambda$ using Eq. 3
9:     Sampling a batch of new transition samples $\mathcal{B}^*$
10:    $\mathcal{D}^* \leftarrow \mathcal{D}^* \cup \mathcal{B}^*$
11: **until** end
12: Train policy $\pi$ on the final dataset $\mathcal{D}^* \cup \mathcal{D}$

---

augment samples with higher returns, we propose three distinct condition selection mechanisms: return-prior, RTG-prior, and random goal conditions. **(1) Return-prior goal condition**. In this approach, we rank all trajectories based on their return values and select the top $n$ trajectories. During the sampling process of the diffusion model, the RTG and timestep pairs $(\hat{g}_t, t)$ from these top $n$ trajectories are selected as the sampling goal conditions. This method filters high-return trajectories from the initial offline datasets and reuses them to sample more well-optimized transitions. **(2) RTG-prior goal condition**. We group RTGs by their associated timesteps and then sort them to select the top $n$ RTGs along with their corresponding timesteps as goal conditions. This approach selectively reuses high-RTG transitions for data augmentation, focusing on transitions that are most likely to yield higher returns. **(3) Random goal condition**. We randomly select $m$ RTG and timestep pairs $(\hat{g}_t, t)$ as sampling goal conditions for each batch of samples. This increases the diversity of the augmented data while paying less attention to the optimal trajectories for improving performance.

### 4.1.3 CONTROLLABLE GOAL SCALING

Selective goal conditions offer high-return guidance during the sampling process but are limited in generating data with returns or quality beyond the initial offline datasets. To overcome this limitation, we introduce a controllable goal scaling factor, $\lambda$, which can be multiplied with the goal values to represent a higher return expectation. This approach enables flexible adjustment of goal values to drive the sampling process toward higher-quality data. As illustrated in Figure 1, a higher RTG goal at each timestep directs the sampling process toward a trajectory with a greater overall return. Since RTG values can be either positive or negative in certain tasks, we propose multiplying positive goals by the scaling factor and dividing negative goals by it.

$$\text{goal} = \begin{cases} (\lambda \hat{g}_t, t), & \hat{g}_t >= 0 \\ (\hat{g}_t/\lambda, t), & \hat{g}_t < 0. \end{cases} \tag{3}$$

### 4.2 ADAPTIVE GATED CONDITIONING

To better capture goal information and integrate conditions with the diffusion model, we propose an adaptive gated conditioning approach, as shown in Figure 2. The conditional inputs include both the noise level condition and goal condition, which are embedded separately, then element-wise added, and fed into the neural network. The noise input is processed with several gated residual multi-layer perception (MLP) blocks with novel adaptive gated skip connections between shallow and deep layers.

**Condition embedding.** The noise level $\sigma$ for diffusion is encoded using Random Fourier Feature Rahimi & Recht (2007) embedding. The RTG is processed with a linear transformation to get a hidden representation. The timestep of each RTG is embedded with Sinusoidal positional embedding Vaswani et al. (2017). We concatenate the RTG and timestep embeddings to form the goal condition, which is then element-wise added to the noise level embedding and used as the conditional input.

**Adaptive gated long skip connection.** As shown in the left part of Figure 2, we adopt a long skip connection similar to U-Net Ronneberger et al. (2015) to connect MLP blocks at different levels. To

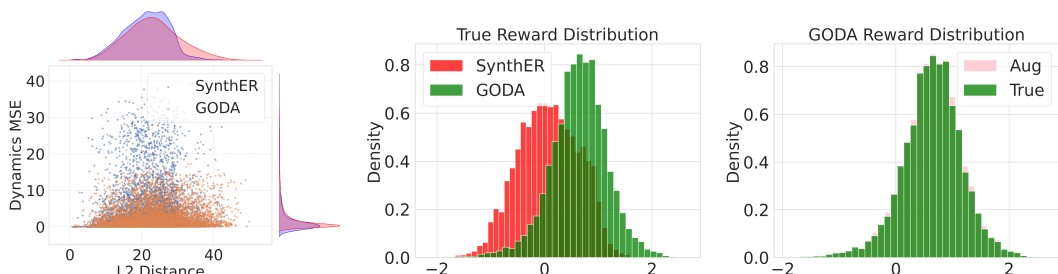

Figure 3: Data quality evaluation for SynthER and GODA on Walker2D-Random-V2. **Left**: Dymanics MSE and L2 Distance comparison. Smaller Dynamics MSE indicates better validity and larger L2 Distance indicates higher diversity. **Middle**: Ground-truth reward distributions from the simulator for augmented datasets. **Right**: Ground-truth and augmented reward distributions for GODA dataset.

Table 1: Data quality evaluation metrics for SynthER and GODA on Walker2D tasks. Smaller Dynamics MSE, larger L2 Distance, and larger Average Reward indicate better quality.

| Task | Dynamics MSE | | L2 Distance | | Average Reward | |
|---|---|---|---|---|---|---|
| | SynthER | GODA | SynthER | GODA | SynthER | GODA |
| Walker2D-Random-v2 | 2.7±5.7 | **1.9±2.9** | 21.8±7.0 | **23.2±7.6** | 0.1±0.6 | **0.6±0.5** |
| Walker2D-Medium-Replay-v2 | 0.5±1.7 | **0.4±1.1** | **17.3±6.3** | 17.2±6.0 | 2.5±1.3 | **3.5±0.9** |
| Walker2D-Medium-v2 | **0.3±1.0** | 0.3±0.8 | 11.7±5.2 | **11.8±5.3** | 3.4±1.2 | **3.7±0.9** |

capture different information with varying importance weights, we propose an adaptive gated long skip connection structure by adding the previous information with an adaptive gate mechanism.

$$x_{\text{out}} = (1 - \omega) * x_{\text{skip}} + \omega * x, \tag{4}$$

where $x_{\text{skip}}$ and $x$ are outputs of a shallower and the previous block, and $\omega$ denotes a learnable weight calculated by regressing the conditional input with an MLP and a sigmoid layer.

**Gated residual MLP block.** The right part of Figure 2 depicts the structure of each gated residual MLP block. We adopt the widely used adaptive layer normalization (adaLN) method Peebles & Xie (2023) to learn dimension-wise scale $\gamma$ and shift $\beta$ based on the conditional information. Besides, we explore a modification of the residual connection He et al. (2016) and introduce a novel adaptive gated residual connection. It also regresses the conditional input and gets a learnable weight $\nu$ for adaptively preserving input information.

$$x_{\text{out}} = (1 - \nu) * F(x) + \nu * x, \tag{5}$$

where $F$ is the learned transformation.

### 4.3 MODEL IMPLEMENTATION

Given the strong ability of diffusion models to capture complex data distribution and generate high-dimension data, we adopt EDM Karras et al. (2022) as our generative model for augmenting offline data. The neural network equipped with adaptive gated conditioning as illustrated in Figure 2 is used as the denoiser function. We train the generative model to approximate the data distribution of the offline dataset and use every transition tuple as a training sample. Given the non-sequence input format, we do not consider complicated structures, e.g., attention mechanisms, but use simple MLPs to process inputs. Algorithm 1 shows the learning process of our GODA method. With goal conditions $\mathbf{c}$ and transition data $\mathbf{x}$ from the original datasets, the generative model $G_\theta$ with a learnable denoiser neural network $D_\theta$ is trained by

$$\min_\theta \mathbb{E}_{\mathbf{x}, \mathbf{c} \sim p_{\text{data}}; \mathbf{n} \sim \mathcal{N}(\mathbf{0}, \sigma^2 \mathbf{I})} \|D_\theta(\mathbf{x} + \mathbf{n}; \sigma; \mathbf{c}) - \mathbf{x}\|_2^2. \tag{6}$$

After obtaining a well-trained conditional diffusion model, we leverage it for sampling data and store data in augmentation dataset $\mathcal{D}^*$ for further policy training.

## 5 EXPERIMENTS

In this section, we provide a comprehensive overview of the experiments conducted to evaluate the performance of our proposed GODA method.

### 5.1 EXPERIMENTAL SETTINGS

**D4RL tasks and datasets.** We adopt three popular Mujoco locomotion tasks from Gym [1], i.e., HalfCheetah, Hopper, and Walker2D, and a navigation task, i.e., Maze2D Fu et al. (2020). In the case of Gym tasks where dense rewards are available, we employ three distinct data configurations from the D4RL datasets: Random, Medium-Replay, and Medium. For Maze2D, a 2D agent is trained to reach a goal position utilizing minimal feedback i.e., a single point for success, zero otherwise. Three datasets collected from different maze layouts are adopted, i.e., Umaze, Medium, and Large. Besides, we further test our GODA on more complex tasks, specifically the Pen and Door tasks from the Adroit benchmark Rajeswaran et al. (2017); Fu et al. (2020). These tasks involve manipulating a pen and opening a door using a 24-DoF simulated hand robot. For each task, we use two different datasets: Human and Cloned. The Human dataset consists of trajectories from human demonstrations, while the Cloned dataset is generated by applying an imitation policy trained from a mix of human and expert demonstrations and combining the resulting trajectories with the demonstrations in a 50/50 split. More details can be found in Appendix B.2.

**Traffic signal control tasks and datasets.** To evaluate GODA's applicability to real-world challenges, we further test it on TSC tasks using the CityFlow simulator Zhang et al. (2019). TSC aims to optimize traffic flow by efficiently managing traffic signals to maximize overall traffic efficiency. As shown in Figure 4, a signalized intersection in TSC problems is composed of approaches with several lanes in each approach. The controller manages the phase as shown in the top right part of Figure 4, which determines the activated traffic signals for different directions, to control the orderly movement of vehicles. To evaluate our GODA, we select three real-world scenarios featuring a 12-intersection grid from Jinan (JN) city and two scenarios with a 16-intersection grid from Hangzhou (HZ) city Zhang & Deng (2023). These scenarios represent a variety of traffic patterns and intersection structures, allowing us to cover a wide range of traffic situations. To bridge the gap between simulation and real-world conditions, we use the widely adopted Fixed-Time (FT) controller as one of our behavior policies for generating the initial offline datasets. Additionally, we employ Advanced Max Pressure (AMP) Zhang et al. (2022) and Advanced CoLight (ACL) Zhang et al. (2022) to create higher-quality datasets for further evaluation. We present more details in Appendix B.3.

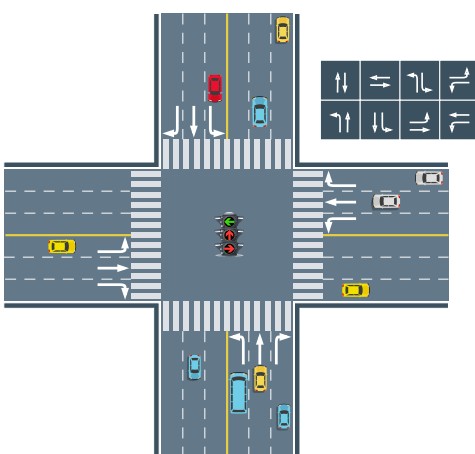

Figure 4: A standard signalized intersection with four three-lane approaches and eight phases.

**Baseline methods.** To verify the effectiveness of our proposed GODA, we compare it with three state-of-the-art data augmentation methods: **TATU** Zhang et al. (2023), which learns world models to generate synthetic rollouts and truncates trajectories with high accumulated uncertainty. **SynthER** Lu et al. (2024), which employs diffusion models to unconditionally augment large amounts of new data based on the learned distribution of original datasets. **DiffStitch** Li et al. (2024), which augments data with a diffusion model and three MLPs, and actively connects low to high-reward trajectories with stitch techniques.

**Evaluation algorithms.** To verify the quality of datasets augmented by GODA, we follow the evaluation settings adopted in previous data augmentation studies. We train two widely-used offline

---

[1]https://www.gymlibrary.dev/environments/mujoco/

Table 2: Normalized scores of GODA and baseline data augmentation methods. The results are calculated across 5 random seeds. Values in bold represent the best performance (largest score).

| Task | Dataset | IQL | | | | | TD3+BC | | | | |
|------|---------|-----|----|----|----|----|----|----|----|----|----|
| | | Original | TATU | SynthER | DStitch | GODA | Original | TATU | SynthER | DStitch | GODA |
| **Halfcheetah** | Rand | 15.2±1.2 | 17.7±2.9 | 17.2±3.4 | 15.8±2.0 | **19.5±0.5** | 11.3±0.8 | 12.1±2.3 | 12.2±1.1 | 11.8±1.4 | **12.5±1.3** |
| | Med-R | 43.5±0.4 | 44.2±0.1 | 46.6±0.2 | 44.7±0.1 | **47.5±0.4** | 44.8±0.7 | 44.5±0.3 | **45.9±0.9** | 44.7±0.3 | 44.9±0.2 |
| | Med | 48.3±0.1 | 48.2±0.1 | 49.6±0.3 | 49.4±0.1 | **50.4±0.1** | 48.1±0.2 | 48.1±0.2 | 49.9±1.2 | **50.4±0.5** | 48.5±0.1 |
| **Walker2D** | Rand | 4.1±0.8 | 6.3±0.5 | 4.2±0.3 | 4.6±1.1 | **14.3±7.1** | 0.6±0.3 | **6.5±4.3** | 2.3±1.9 | 2.4±1.0 | 4.2±1.8 |
| | Med-R | 82.6±8.0 | 75.0±12.1 | 83.3±5.9 | 86.6±2.8 | **96.1±4.9** | 85.6±4.6 | 62.1±10.4 | 90.5±4.3 | 89.7±4.2 | **93.0±5.6** |
| | Med | 84.0±5.4 | 76.6±10.7 | **84.7±5.5** | 83.2±2.2 | 79.1±2.4 | 82.7±5.5 | 75.8±3.5 | 84.8±1.4 | 83.4±1.7 | **86.2±0.7** |
| **Hopper** | Rand | 7.2±0.2 | 8.1±2.9 | 7.7±0.1 | 6.5±0.9 | **8.7±2.1** | 8.6±0.3 | **18.1±11.5** | 14.6±9.4 | 8.8±2.3 | 8.2±0.1 |
| | Med-R | 84.6±13.5 | 79.6±7.6 | **103.2±0.4** | 102.1±0.4 | 102.5±0.6 | 64.4±24.8 | 64.1±10.5 | 53.4±15.5 | **79.6±13.5** | 63.0±12.8 |
| | Med | 62.8±6.0 | 60.3±3.6 | 72.0±4.5 | 71.0±4.2 | **74.3±2.9** | 60.4±4.0 | 58.3±4.8 | 63.4±4.2 | 60.3±4.9 | **74.8±3.6** |
| **Average** | | 48.0±4.4 | 46.2±4.3 | 52.1±2.4 | 51.5±1.5 | **54.7±2.2** | 45.2±7.4 | 43.3±4.2 | 46.3±4.7 | 47.9±3.3 | **48.4±3.9** |
| **Maze2D** | Umaze | 37.7±2.0 | 33.0±4.8 | 41.0±0.7 | 38.5±6.2 | **59.5±2.6** | 29.4±14.2 | 37.7±10.9 | 37.6±14.4 | 38.4±7.5 | **46.4±8.3** |
| | Med | 35.5±1.0 | 35.1±1.3 | 35.1±2.6 | 35.5±1.5 | **35.8±2.6** | 59.5±41.9 | 73.8±36.9 | 65.2±36.1 | 66.8±30.9 | **86.5±26.4** |
| | Large | 49.6±22.0 | 69.1±20.1 | 60.8±5.3 | 68.4±12.6 | **109±16.5** | 97.1±29.3 | 93.1±25.3 | 92.5±38.5 | 92.4±36.2 | **104.3±20.1** |
| **Average** | | 40.9±8.3 | 45.7±8.2 | 45.6±2.9 | 47.5±6.8 | **68.1±6.6** | 62.0±28. | 68.2±10. | 65.1±29. | 65.9±24.9 | **79.1±7.5** |

RL algorithms, i.e., IQL Kostrikov et al. (2021) and TD3+BC Fujimoto & Gu (2021), on datasets and evaluate the learned policy on D4RL tasks. For TSC tasks, we utilize BCQ Fujimoto et al. (2019), CQL Kumar et al. (2020), and DataLight Zhang & Deng (2023) as the evaluation algorithms.

We augment 5M samples for each D4RL task and 200K samples for each TSC task. It is important to note that for GODA, we train the evaluation algorithms using a mix of the original datasets and the augmented datasets, whereas for the other baseline methods, only the augmented datasets are used. This is because GODA focuses on augmenting samples from the high-reward zones, which may lead to reduced data diversity. In contrast, the baseline methods, as reported in their respective papers Lu et al. (2024) and our experiments, exhibit degraded or similar performance when using a mix of the original and augmented datasets.

## 5.2 DATA QUALITY MEASUREMENT

Since our GODA is built upon SynthER, we compare the quality of the datasets augmented by both SynthER and GODA to assess whether the goal conditions incorporated by GODA enhance data quality. We adopt two metrics from SynthER Lu et al. (2024), i.e., Dynamics MSE $= \frac{1}{M} \sum_{i=1}^{M} \left( (s_{t+1}^i, r_t^i) - (\hat{s}_{t+1}^i, \hat{r}_t^i) \right)^2$ and L2 Distance $= \frac{1}{M} \sum_{i=1}^{M} \left( (s_t^i, a_t^i) - (\bar{s}_t^i, \bar{a}_t^i) \right)^2$, and introduce the Average Reward $= \frac{1}{M} \sum_{i=1}^{M} \hat{r}_t^i$, where $M$ is the selected number of samples, $s_t^i, a_t^i, s_{t+1}^i$, $r_t^i$ denote the samples from augmented datasets, $\hat{s}_{t+1}^i$ and $\hat{r}_t^i$ denote the next state and reward generated by the simulator given states and actions from augmented datasets, and $\bar{s}_t^i$ and $\bar{a}_t^i$ are the state and action from original datasets. Dynamics MSE measures how well the augmentation models capture the dynamics of the environment by learning patterns from the original datasets and generating data that aligns with those dynamics. L2 Distance assesses the models' exploration capabilities and data diversity by calculating the Euclidean distance between the augmented dataset and the original dataset, reflecting how diverse the generated data is. Average Reward compares the ground-truth reward distributions produced by the simulator given states and actions in datasets augmented by SynthER and GODA.

The left part of Figure 3 presents a scatter plot of 10K points sampled from the augmented datasets. Results show that datasets generated by GODA exhibit much lower Dynamics MSE and a wider range of L2 Distance values, indicating both better alignment with environmental dynamics and greater diversity. The superior validity in performance may stem from the goal conditions (RTG-timestep pairs), which provide critical information for generating samples that better match the environment's dynamics. Meanwhile, the increased diversity is likely due to the scaled out-of-distribution goal conditions incorporated in the sampling process. The middle part demonstrates that GODA not only generates samples within a high-reward data zone but also extends the boundary of high rewards beyond the best demonstrations, compared to SynthER. The right part shows that the rewards generated by GODA align closely with the ground-truth values. The evaluation results for the three metrics in Table 1 further demonstrate that GODA outperforms SynthER in terms of all data quality evaluation metrics across nearly all Walker2D tasks. Further detailed evaluation results can be found in Appendix C.1.

Table 4: Average travel time comparison on real-world TSC tasks. Smaller travel time indicates better traffic efficiency.

| Traffic | Dataset | BCQ | | | CQL | | | DataLight | | |
|---|---|---|---|---|---|---|---|---|---|---|
| | | Original | SynthER | GODA | Original | SynthER | GODA | Original | SynthER | GODA |
| JN 1 | FT | 269.7±4.1 | 267.9±2.7 | **264.1±4.4** | 272.0±2.1 | 273.4±2.5 | **271.7±4.5** | 279.8±2.9 | 274.1±1.6 | **270.7±4.1** |
| | AMP | 271.5±3.9 | 264.0±6.1 | **259.7±0.9** | 261.8±0.3 | 261.7±4.3 | **260.6±3.4** | **298.1±3.1** | 299.2±2.0 | 301.7±1.4 |
| | ACL | 271.1±2.4 | 271.9±1.4 | **270.6±0.3** | 273.3±1.6 | 275.2±4.4 | **273.2±4.3** | 256.4±3.2 | 258.4±2.5 | **255.3±0.3** |
| JN 2 | FT | 267.2±3.6 | **265.5±1.2** | 266.6±5.1 | **269.3±0.3** | 275.3±6.9 | 272.5±1.9 | 293.9±1.7 | 288.2±2.2 | **281.0±0.3** |
| | AMP | **250.7±0.7** | 254.7±2.6 | 252.1±3.4 | 251.9±4.0 | 249.4±5.3 | **245.5±1.0** | 244.4±2.3 | **240.0±2.2** | 240.9±4.0 |
| | ACL | **253.4±0.3** | 265.7±1.5 | 262.0±4.5 | 248.1±0.3 | 248.2±2.1 | **248.0±2.0** | 241.9±0.3 | 236.4±0.3 | **235.1±0.5** |
| JN 3 | FT | 266.9±3.6 | 263.5±4.9 | **257.3±3.4** | 268.0±0.7 | 273.7±2.3 | **267.1±2.8** | 302.6±1.9 | 299.9±5.0 | **299.8±1.8** |
| | AMP | 263.8±3.1 | 259.3±0.7 | **253.2±4.4** | 251.5±2.9 | 247.7±4.9 | **242.5±3.5** | 239.4±1.8 | 241.8±4.9 | **232.5±1.7** |
| | ACL | **242.2±4.0** | 245.7±0.6 | 243.2±1.6 | **242.1±1.4** | 244.5±4.0 | 245.5±7.1 | 240.3±3.6 | 234.7±1.3 | **230.1±2.7** |
| Average | | 267.9±3.2 | 267.1±2.3 | **263.9±2.7** | 264.8±1.9 | 265.7±4.2 | **262.4±3.2** | 267.5±2.1 | 265.5±2.3 | **262.6±1.4** |
| HZ 1 | FT | 324.5±7.3 | 313.2±2.0 | **310.5±0.8** | 317.4±5.7 | 315.5±4.0 | **307.1±2.7** | 290.1±0.6 | 290.8±0.3 | **287.2±0.3** |
| | AMP | **295.8±4.6** | 302.7±1.9 | 301.7±4.1 | 300.0±0.3 | 295.1±0.3 | **285.4±1.0** | 287.3±4.3 | **284.9±3.8** | 286.1±3.2 |
| | ACL | 281.7±1.1 | 281.3±6.5 | **281.1±1.3** | 288.6±3.3 | 286.3±2.8 | **278.9±2.1** | 284.9±5.6 | 282.1±1.7 | **278.0±3.8** |
| HZ 2 | FT | **340.0±4.9** | 340.7±4.6 | 341.3±1.5 | 341.7±2.7 | 334.8±1.4 | **331.6±1.9** | 308.0±0.3 | **308.4±3.1** | 309.2±3.0 |
| | AMP | 332.5±0.3 | 324.8±3.0 | **316.9±1.9** | **318.0±0.6** | 321.9±3.3 | 321.4±3.7 | 312.3±3.6 | 310.4±2.2 | **308.4±2.2** |
| | ACL | 336.7±1.9 | 329.3±0.5 | **327.1±2.8** | 347.4±3.8 | 343.9±2.6 | **336.8±0.3** | 317.6±4.1 | **314.8±4.1** | 315.4±3.0 |
| Average | | 317.3±2.6 | 315.8±3.3 | **313.6±2.3** | 319.1±2.1 | 316.4±2.1 | **310.8±1.8** | 302.0±3.6 | 300.1±3.0 | **299.4±3.0** |

## 5.3 Performance on D4RL

**Gym tasks.** Table 1 presents a performance comparison between GODA and other state-of-the-art data augmentation methods trained on the D4RL Gym and Maze2D. We adopt the results of Original and SyntheER from the SynthER paper Lu et al. (2024), and those of TATU and DStitch from the DStitch Li et al. (2024) paper. We further conduct experiments for tasks not covered in the literature. As shown in the results, GODA consistently outperforms other methods across most Gym locomotion tasks when evaluated with both IQL and TD3+BC, resulting in higher average scores. Notably, even for tasks using Random datasets, GODA successfully leverages limited high-quality samples to enhance data quality, leading to improved final performance.

**Maze2D tasks.** For Maze2D tasks where rewards are sparse, GODA demonstrates significant improvements across all datasets, achieving average gains of 43.4% and 16.0% over the best baseline methods when evaluated with IQL and TD3+BC, respectively. The maximum improvement reaches 57.7% when applying GODA to the Maze2D Large dataset. This highlights GODA's ability to effectively capture data distributions of various types of datasets and consistently augment high-quality samples.

**Adroit tasks.** Table 3 presents the normalized scores on Adroit Pena and Door tasks, evaluated using the IQL algorithm, as TD3+BC fails to perform on these tasks. As shown, GODA outperforms all baselines on the Pen-Cloned dataset, although it underperforms on Pen-Human.

Table 3: Normalized scores of GODA and baseline data augmentation methods on Adroit tasks evaluated using IQL.

| Task | Dataset | Original | TATU | SynthER | DStitch | GODA |
|---|---|---|---|---|---|---|
| Pen | Human | 79.1±28.5 | 88.9±22.6 | **96.8±8.6** | 87.4±8.6 | 75.6±31.4 |
| | Cloned | 45.8±29.9 | 52.5±27.9 | 45.3±23.4 | 64.0±29.6 | **64.8±20.6** |
| Average | | 62.4±29.2 | 70.7±25.2 | 71.0±16.0 | **75.7±19.1** | 70.2±26.0 |
| Door | Human | 1.6±2.1 | 7.0±1.6 | 8.3±2.2 | 10.0±2.5 | **14.8±5.0** |
| | Cloned | -0.1±0.5 | -0.1±0.3 | 5.9±1.8 | 4.4±0.4 | **16.8±6.1** |
| Average | | 0.8±1.3 | 3.5±1.0 | 7.1±2.0 | 7.2±1.4 | **15.8±5.5** |

However, for both datasets in the Door task, GODA demonstrates significant improvements over the best baseline methods. These results further demonstrate that GODA is capable of handling more complex tasks.

## 5.4 Extended Experiments on Traffic Signal Control

Table 4 presents a comparison of average travel times across different methods for TSC tasks. As shown, while SynthER achieves modest improvements over the performance of models trained on the original datasets, it fails to surpass the original datasets on JN tasks when using the CQL algorithm. In contrast, our GODA consistently outperforms both the original datasets and SynthER across most tasks and all average evaluations. These extended experiments on TSC tasks further val-

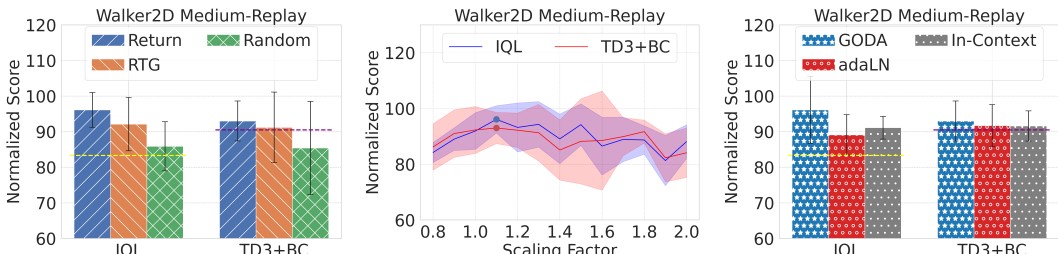

Figure 5: Ablation studies on condition selection mechanism, goal scaling factor, and adaptive gated conditioning from left to right. Yellow and purple horizon lines represent results for SynthER.

idate that GODA is not only applicable to diverse tasks but also capable of improving performance in real-world challenges.

## 5.5 ABLATION STUDY

To validate the effectiveness of GODA's components, we conduct experiments using different configurations. More detailed ablation and sensitivity studies can be found in Appendix C.

**Condition Selection.** We test three condition selection mechanisms as described in Section 4.1.2: return-prior, RTG-prior, and random goal conditions. As shown in the left part of Figure 5, the return-prior method demonstrates superior performance compared to the other two approaches. Moreover, GODA with the return- and RTG-prior conditions outperforms SynthER when tested on two offline RL algorithms. In contrast, the random-prior method shows results comparable to SynthER. This suggests that high-goal conditions identified by the return- and RTG-prior methods enable GODA to generate samples beyond the original data distribution. Randomly selected goal conditions, however, fail to target high-reward regions, producing similar results to SynthER.

**Goal scaling factor.** We further examine the effect of different scaling factors, testing values ranging from 0.8 to 2.0. As seen in the middle part of Figure 5, the performance improves as the scaling factor increases but slightly degrades when the scaling factor exceeds 1.1. Scaling factors below 1.0 shrink the selected goals, leading to suboptimal samples. Conversely, scaling factors above 1.1 push the selected goals too far beyond the training data distribution, resulting in diminished performance.

**Adaptive gated conditioning.** Finally, we evaluate the impact of the adaptive gated conditioning method. We compare GODA with two variants: one using only adaLN conditioning Peebles & Xie (2023), and another using in-context conditioning, where condition embeddings are directly appended to the input embeddings. From the right part of Figure 5, it is clear that GODA with adaptive gated conditioning achieves the best results and the adaLN and in-context conditioning show similar performance. Additionally, all three methods outperform SynthER which lacks goal conditions. This demonstrates that the inclusion of goal conditions is crucial for guiding the sampling process toward high returns, and our adaptive gated conditioning method enhances the model's ability to fully utilize these conditions.

## 6 CONCLUSION AND DISCUSSION

This paper proposes a novel goal-conditioned data augmentation method, namely GODA, which integrates goal guidance into the data augmentation process. We define the easily obtainable return-to-go signal, along with its corresponding timestep in a trajectory, as the goal condition. To achieve high-return augmentation, we introduce several goal selection mechanisms and a scaling method. Additionally, we propose a novel adaptive gated conditioning structure to better incorporate goal conditions into our diffusion model. We demonstrate that data augmented by GODA shows higher quality than SynthER without goal conditions on different evaluation metrics. Extensive experiments on MuJoCo locomotion tasks and a maze task confirm that GODA enhances the performance of classic offline RL methods when trained on GODA-augmented datasets. Furthermore, we evaluate GODA on real-world traffic signal control tasks. The results demonstrate that GODA is highly applicable to TSC problems, making RL-based methods more practical for real-world applications.

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

Figure 6: An illustrative example of how GODA utilizes higher goals to steer the sampling process toward higher-reward data distribution region.

## A HYPERPARAMETERS

In this section, we show more details about the hyperparameter settings of the GODA model.

### A.1 DENOISING NETWORK

The denoising neural network utilizes the adaptive gated conditioning architecture, as shown in Figure 2. Table 5 details the associated hyperparameters. We use Random Fourier Feature embedding Rahimi & Recht (2007) and Sinusoidal positional embedding Vaswani et al. (2017) to process the noise level and timestep of each RTG respectively, with an embedding dimension of 128. The width of the linear layers in the MLP block is set to 512, with SiLU Elfwing et al. (2018) as the activation function. The total number of trainable parameters for the denoiser neural network is approximately 3.3M. We train our GODA model with 100K steps of gradient updates, with a batch size of 256 and a learning rate of 0.0003.

Table 5: Hyperparameter settings for denoiser network.

| Hyperparameter | Value |
|---|---|
| embedding dimension | 128 |
| MLP width | 512 |
| MLP activation | SiLU |
| gate activation | Sigmoid |
| learning rate | 0.0003 |
| batch size | 256 |
| learning rate schedule | cosine annealing |
| optimizer | Adam |
| gradient update steps | 100K |

### A.2 ELUCIDATED DIFFUSION MODEL

We adopt EDM Karras et al. (2022) as our diffusion model and follow the original settings from SynthER Lu et al. (2024), with the default hyperparameters shown in Table 6. EDM employs Heun's $2^{nd}$ order ODE solver Ascher & Petzold (1998) to solve the reverse-time ODE, enabling data sampling through the reverse process. The diffusion timestep is set to 128 for higher-quality results. All training and sampling are conducted on an AMD Ryzen 7 7700X 8-Core Processor and a single NVIDIA GeForce RTX 4080 GPU. Training GODA for 100K steps takes approximately 14 minutes while generating 5M samples with a sampling batch size of 250K requires about 300 seconds.

Table 6: Hyperparameter settings for the diffusion model.

| Hyperparameter | Value |
|---|---|
| number of diffusion steps | 128 |
| $S_{\text{churn}}$ | 80 |
| $S_{\text{tmin}}$ | 0.05 |
| $S_{\text{tmax}}$ | 50 |
| $S_{\text{noise}}$ | 1.003 |
| $\sigma_{\text{min}}$ | 0.002 |
| $\sigma_{\text{max}}$ | 80 |

## B    EXPERIMENT DETAILS

### B.1    BASELINE AND EVALUATION RL METHODS

For data augmentation baseline methods, we adopt the implementation of TATU from https://github.com/pipixiaqishi1/TATU, DiffStitch from https://github.com/guangheli12/DiffStitch, and SynthER from https://github.com/conglu1997/SynthER to conduct experiments not covered in their respective papers

For the evaluation RL methods, we use IQL and TD3+BC from the Clean Offline Reinforcement Learning (CORL) codebase Tarasov et al. (2024) for D4RL tasks. For TSC tasks, we employ the implementation of BCQ, CQL, and DataLight from https://github.com/LiangZhang1996/DataLight.

### B.2    D4RL SETTINGS

For Gym-MuJoCo tasks from the D4RL benchmark, Random, Medium-Replay, and Medium datasets are adopted. To elaborate, Random datasets contain transition data generated by a randomly initialized policy. Medium datasets consist of a million data points gathered using a policy that achieves one-third of the performance of an expert approach. Medium-replay datasets contain the stored experience in a replay buffer during the training of a policy until it reaches the score in Medium datasets.

### B.3    TSC SETTINGS

We further elaborate on the TSC tasks adopted as the real-world challenges in this section. We formulate the TSC problem as a MDP and define the state, action, and reward function as follows:

**State.** For behavior policies (AMP and ACL), the state representation includes the current phase, traffic movement efficiency pressure, and the number of effective running vehicles Zhang et al. (2022). For evaluation algorithms, BCQ and CQL use the same state representation as AMP, while DataLight adopts the number of vehicles, along with the total velocity saturation and unsaturation degrees Zhang & Deng (2023).

**Action.** The action is generally defined as the phase selection for the next time period.

**Reward.** AMP, BCQ, CQL and DataLight use pressure Zhang et al. (2022) as the reward while ACL uses queue length. It is worth noting that we use the opposite of these metrics as the final reward function.

Due to the absence of certain key signals in the original datasets from the DataLight codebase, we generate a total of 24K samples for each dataset using the behavior policies for each task. Additionally, we augment 20K samples for each task using our GODA model. The task horizon for each TSC scenario is set to 3600 seconds, with a control step of 15 seconds. For the sampling process, we employ the return-prior goal condition selection method and set the goal scaling factor to 0.85, given the negative reward values in TSC tasks.

## C    FURTHER EXPERIMENTAL RESULTS

In this section, we show some more experimental results for our GODA model.

### C.1    DATA QUALITY EVALUATION

As shown in Figures 7 and 8, the datasets augmented by GODA exhibit lower dynamics MSE across all Walker2D tasks, indicating better alignment with the environment's dynamics. For L2 Distance, GODA outperforms SynthER on two tasks while delivering comparable results on Walker2D-Medium-Replay-v2. In terms of Average Reward, GODA consistently augments data in the high-reward zone, extending rewards beyond the original data distribution to even higher values.

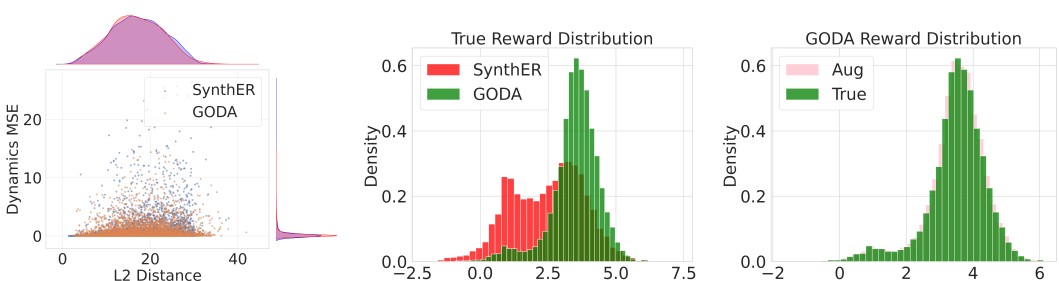

Figure 7: Data quality evaluation for SynthER and GODA on Walker2D-Medium-Replay-V2. **Left**: Dymanics MSE and L2 Distance comparison. Smaller Dynamics MSE indicates better validity and larger L2 Distance indicates higher diversity. **Middle**: Ground-truth reward distributions from the simulator for augmented datasets. **Right**: Ground-truth and augmented reward distributions for GODA dataset.

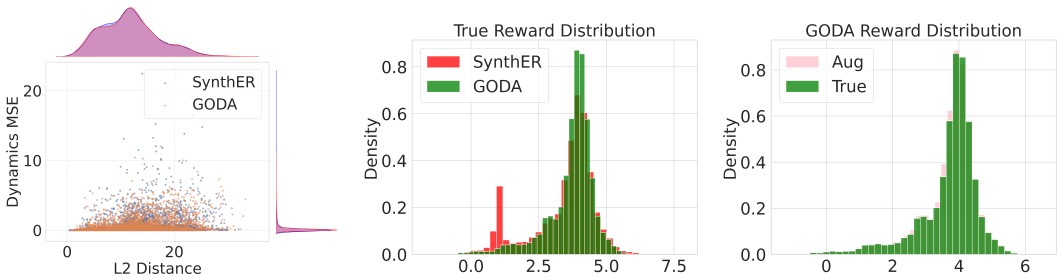

Figure 8: Data quality evaluation for SynthER and GODA on Walker2D-Medium-V2.

### C.2    ABLATION ON TOP CONDITIONS SELECTION

Given that the original datasets contain varying numbers of trajectories, the number of top conditions selected for sampling may differ across datasets. We compare different selections of the top $n$ conditions for each dataset. Based on the results shown in Figure 9, we empirically select 50 top conditions for the Random and Medium-Replay datasets, and 40 for the Medium datasets.

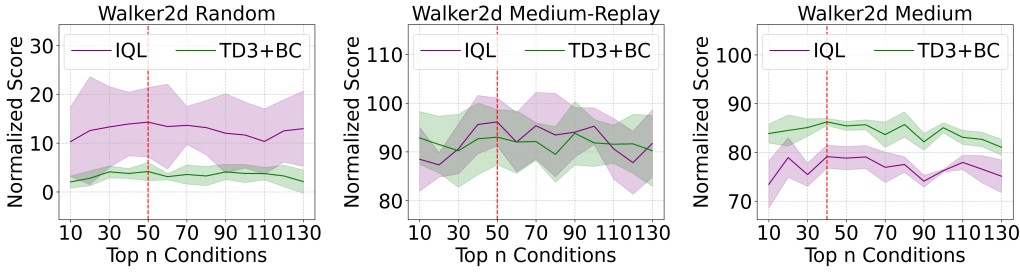

Figure 9: Ablation study on top n conditions.

## C.3 ABLATION ON MIXED DATASETS

Since GODA primarily augments samples from high-reward regions of the data distribution, which might result in a lack of diversity, we use a mix of both the original and augmented datasets for training. In this section, we compare the performance of our default setting (mixed datasets) with the use of only augmented datasets. As shown in Figures 10 and 11, removing the original datasets leads to slight performance degradation across most tasks. Therefore, combining our augmented datasets with the original datasets helps increase the diversity and extend the reward boundary.

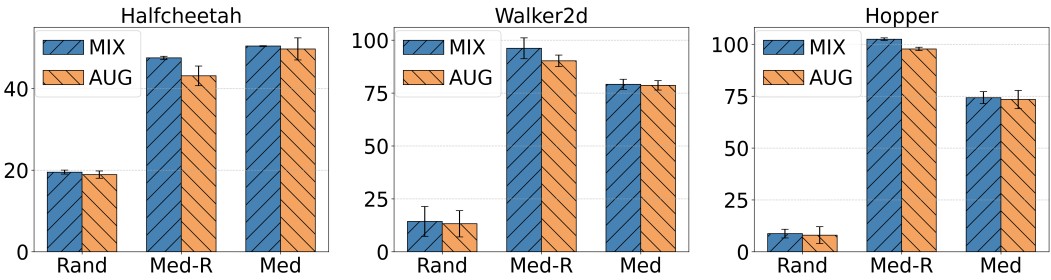

Figure 10: Ablation on mixed datasets for IQL evaluation.

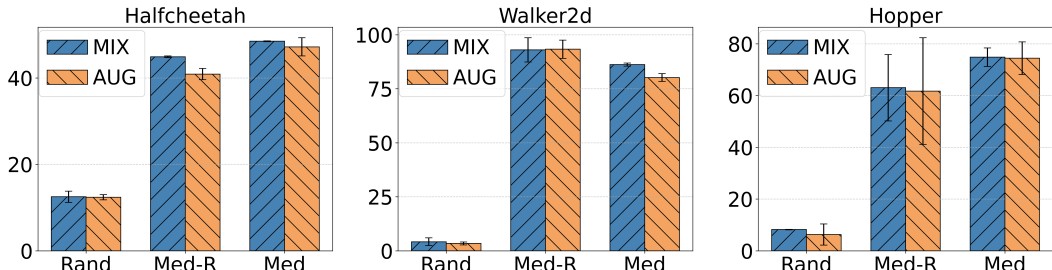

Figure 11: Ablation on mixed datasets for TD3+BC evaluation.

