# OpenReview forum: "GODA: Goal-conditioned Data Augmentation"
_ICLR.cc/2025/Conference — ICLR 2025 Conference Withdrawn Submission_

### Official Review · Reviewer_TR5m · 2024-10-24

**Soundness:** 2
**Presentation:** 2
**Contribution:** 2
**Rating:** 3
**Confidence:** 4

**Summary:**

This paper presents a method that augments offline dataset with high-quality demonstrations by learning a conditional diffusion model and guided sampling. Authors conduct experiments on D4RL and TSC tasks with different offline RL algorithms to verify the effectiveness of the method.

**Strengths:**

In offline RL, limited and sub-optimal dataset leads to various problems. Augmenting dataset with generative model which properly learn the underlying transition dynamics might be helpful for solving offline RL problems. Furthermore, authors extend their algorithm into TSC tasks, which is not widely used for evaluating offline RL algorithms (usually D4RL is used over 4 years…) but practical scenarios.

**Weaknesses:**

**Abuse of goal-conditioning)**

There are a line of works that study goal-conditioned RL (GCRL, [1]), and typically goal is defined as a vector or a set of vectors of the desired locations. I find that the goal indicates return in this paper. It makes readers very confusing. I strongly recommend to adjust goal-conditioning to return-conditioning.

**Generate transitions by conditioning on return)**

It seems that GODA generates transitions conditioned on the cumulative future rewards for the current state. It is unnatural since cumulative future rewards is an unbiased but very high-variance estimator for the expected return. In my humble opinion, it is really hard to generate a transition that yields high return. Successful generative models for offline RL by return-conditioning also generate trajectories [2, 3]

**Similar Prior works)**

There is a prior work called GTA [4], which also augment dataset for offline RL problems. It seems that the algorithm is very similar in terms of classifier-free guidance, scaling factor for conditioning. The main difference is that GTA generates trajectories and partial noising framework. Could authors verify the contribution of the paper?

[1] Liu, Minghuan, Menghui Zhu, and Weinan Zhang. "Goal-conditioned reinforcement learning: Problems and solutions." *arXiv preprint arXiv:2201.08299* (2022).

[2] Janner, Michael, et al. "Planning with Diffusion for Flexible Behavior Synthesis." *International Conference on Machine Learning*. PMLR, 2022.

[3] Chen, Lili, et al. "Decision transformer: Reinforcement learning via sequence modeling." *Advances in neural information processing systems* 34 (2021): 15084-15097.

[4] Lee, Jaewoo, et al. "GTA: Generative Trajectory Augmentation with Guidance for Offline Reinforcement Learning." *Advances in Neural Information Processing Systems* 37

**Questions:**

**Hyperparameter Sensitivity)**

It seems that the hyperparameter $\lambda$, which controls the guidance level. Does the hyperparameter is tuned for different environment? I cannot find information about $\lambda$.

**TSC Environment)**

It seems that the performance differences is not big across algorithms. Is it natural case?

**Random goal condition)**

In random goal condition, authors say it increases diversity while trading-off quality but I cannot find any validation. Could authors verify that random goal condition increases diversity?

---

### Official Review · Reviewer_CmcJ · 2024-11-03

**Soundness:** 3
**Presentation:** 2
**Contribution:** 2
**Rating:** 5
**Confidence:** 5

**Summary:**

The study introduces GODA (Goal-conditioned Data Augmentation), a diffusion-based methodology aimed at improving data quality for offline reinforcement learning (RL) in environments with scarce optimal demonstrations. GODA employs return-oriented goal conditioning to direct data sampling towards high-return objectives, hence enhancing the utility of suboptimal datasets. The method presents a modifiable scaling strategy to regulate goal intensity and an adaptive gated conditioning mechanism for efficient goal representation. GODA exhibits enhanced data quality and performance on the D4RL benchmark and real-world traffic signal management tasks, surpassing alternative data augmentation techniques in offline reinforcement learning.

**Strengths:**

1. GODA presents an innovative integration of goal conditioning and diffusion models for data augmentation in offline reinforcement learning, employing return-to-go (RTG) to direct high-quality data sampling.

2. The method includes things like adaptive gated conditioning and adjustable objective scaling, which make the data more accurate and varied, as shown in tests on D4RL and traffic signal control tasks.

3. Through the creation of synthetic data for offline reinforcement learning, GODA enhances the diversity and quality of data, allowing the community to use syntehtic data without producing an excessive amount of it.

4.Essential features are clearly explained in the study, however some technical aspects, like adaptive gated conditioning, need more explanation.

**Weaknesses:**

1. Restricted Relevance to Visual Offline Reinforcement Learning: The absence of experiments on visual, image-based tasks constrains GODA's usefulness in real-world scenarios, as several offline reinforcement learning situations involve visual input. The method's application to visual tasks would significantly increase its efficacy and impact on wider reinforcement learning applications.

2. Absence of Data Efficiency Investigation: The research does not address whether GODA's methodology significantly reduces the necessity for larger datasets or extensive data generation. Discussing data efficiency would illustrate the efficacy of GODA's synthetic data generation technique, potentially by comparing performance across various dataset sizes.

**Questions:**

1. Restricted Relevance to Visual Offline Reinforcement Learning: The absence of trials on visual, image-based tasks constrains GODA's usefulness in real-world scenarios, as several offline reinforcement learning contexts require visual data. The method's efficacy and impact on broader RL applications would be significantly enhanced by its application to visual tasks.

2. Lack of Examination of Data Efficiency: The research does not address whether GODA's methodology significantly reduces the necessity for larger datasets or extensive data generation. Discussing data efficiency would illustrate the efficacy of GODA's synthetic data generation technique, potentially by comparing performance across various dataset sizes.

---

### Official Review · Reviewer_NN4F · 2024-11-03

**Soundness:** 3
**Presentation:** 2
**Contribution:** 2
**Rating:** 3
**Confidence:** 4

**Summary:**

This paper introduces Goal-conditioned Data Augmentation (GODA) for offline reinforcement learning (RL). GODA leverages a goal-conditioned diffusion model, which is fundamentally a reward-to-go-conditioned model designed to generate synthetic data with higher rewards. GODA incorporates several structural features in the reverse diffusion process: goal condition selection, controllable goal scaling, and adaptive gated conditioning, enhancing the generation of high-return trajectories. The authors compare GODA against several state-of-the-art data augmentation techniques in offline RL across various benchmarks, including D4RL and Traffic Signal Control (TSC) tasks. Their experiments highlight the effectiveness of GODA in improving data quality and achieving superior performance over other methods.

**Strengths:**

- The paper writing is generally well-structured and comprehensive.
- The experiment is comprehensive with diverse D4RL tasks and real-world examples in TSC.
- The performance outperforms some of the SOTAs, e.g. Synther.

**Weaknesses:**

While the paper presents innovative elements, it has notable weaknesses in problem framing, literature discussion, and clarity in methodology. These limitations reduce the clarity of the authors' contributions. Key issues include:

1. **Related Work Gaps**: Although GODA focuses on relabeling reward-to-go goals during sampling, the paper lacks discussion on established hindsight relabeling methods and generative trajectory augmentation techniques (e.g., GTA [1]). Moreover, in long-horizon goal-conditioned tasks, leveraging hindsight information is a common approach (e.g., [2]), yet these studies are overlooked in the literature review.

2. **Incremental Contribution**: GODA extends previous methods, such as SynthER, by adding controllable reward scaling and adaptive gated conditioning to the model’s inputs, which appears incremental given prior work in the field (e.g., [1]). The paper could benefit from additional experimental comparisons with hindsight relabeling methods for offline RL, such as [2].

3. **Methodology Clarity**: The presentation of GODA’s methodology is insufficiently detailed. The rationale for proposing three distinct condition selection mechanisms is unclear. Furthermore, the controllable goal scaling lacks any systematic approach to hyperparameter $\lambda$ selection, while the conditioning structure has been previously explored (e.g., [4]).

4. **Experimental Design**: Most experiments are conducted in dense-reward settings, deviating from traditional goal-conditioned scenarios that feature sparse rewards.

5. **Minor Notational Issues**: Standard notations, such as $\varepsilon$ for noise in diffusion models, are replaced with $\mathbf{n}$ in this paper, potentially causing confusion. Additionally, notation conventions (e.g., L2 distance and dynamics MSE) should adhere to norms like $\|\|\cdot\|\|_2^2$ rather than $(\cdot)^2$ (see page 8, line 410).

> [1] Lee, Jaewoo, et al. "GTA: Generative Trajectory Augmentation with Guidance for Offline Reinforcement Learning." NeurIPS 2024.
>
> [2] Furuta, Hiroki, et al. "Generalized decision transformer for offline hindsight information matching." ICLR 2022.
>
> [3] Liu, Hao, and Pieter Abbeel. "Emergent agentic transformer from chain of hindsight experience." ICML 2023.
>
> [4] Kumar, Aviral, Xue Bin Peng, and Sergey Levine. "Reward-conditioned policies." *arXiv preprint arXiv:1912.13465* (2019).

**Questions:**

1. **Clarification on Problem Setting**: What specific goal is addressed in GODA’s context? Is it merely the returns-to-go?
2. **Clarification on Figure 2**: Other than the use of adaLN, are there any distinct architectural differences from existing models?
3. **Definition of L2 Distance and Dynamics MSE**: Does the L2 distance refer to the variance between states and actions? If so, could this notation be clarified to avoid misunderstandings?
4. **Intuition Behind $\lambda$ Parameter**: Are there guidelines or benchmarks for tuning $\lambda$ in GODA’s goal scaling? How should adjustments be made to optimize model performance based on synthetic data?
5. **Experiment Setting**: The authors mention using mixed datasets for GODA evaluations, while baselines use only augmented datasets. This is only empirically justified by SynthER’s results. Could the authors provide ablation studies on varying the proportions of mixed synthetic data?

---

### Official Review · Reviewer_2pko · 2024-11-04

**Soundness:** 2
**Presentation:** 3
**Contribution:** 3
**Rating:** 5
**Confidence:** 4

**Summary:**

This work extends SynthER to a goal-conditioned data augmentation for offlineRL setting using return and RTG conditioning information and a conditioning mechanism. Reported evaluations demonstrate the proposed goal-conditioned diffusion model achieves better performance in offlineRL tasks on two different offline policies compared to baselines.

**Strengths:**

- Using conditioning, especially return-based conditions make a lot of sense for generating samples for offlineRL.
- Evaluation is done thoroughly and on various datasets with different level of trajectory quality and offline policies, demonstrating the effectiveness of the conditional generation

**Weaknesses:**

- There is a mismatch in terms of evaluation of the proposed method compared to all baselines which may cause discrepancies and needs to be ablated. It appears that for all baselines, offline policies are only trained on generated samples and the original data was not used. Only in the setting with proposed method, the original data is mixed with generated data. This is a considerable and notable difference between baselines and proposed method that makes the reported evaluations unfair to baselines. Although there is a reference given for why this decision is made, it is necessary to add additional experiments, show how all methods perform when 100% real and x% generated data is used. There should be also experiments to show how performance and percentage of additional generated data are related. In realistic scenarios, if one has access to an offline dataset to train a generative model, they will definately would want to use the original data to maximise their performance.

- This work's novelty is limited due to only extending SynthER to a goal conditioned setting, though if fair evaluations as mentioned above are provided and still the competitiveness holds, this is a less important matter.

**Questions:**

see weaknesses section.

---

### Note · Authors · 2024-11-27

**Comment:**

Dear Editor and Reviewers,

Thank you for the valuable comments on our paper. We deeply appreciate the time and effort you dedicated to reviewing our work, which has provided significant insights for improving our research. After careful consideration and discussion among all authors, we have decided to withdraw the paper at this time. We aim to address the reviewers’ comments thoroughly in our future revisions to ensure the paper meets the highest standards.

Thank you once again for your time and understanding.

**Withdrawal Confirmation:**

I have read and agree with the venue's withdrawal policy on behalf of myself and my co-authors.